# A Comparison of Negative Pressure and Conventional Therapy in Spine Infections: A Single-Center Retrospective Study

**DOI:** 10.3390/jpm13020162

**Published:** 2023-01-17

**Authors:** Wenqiang Xing, Yang Yang, Yun Bai, Xiuchun Yu, Zhengqi Chang

**Affiliations:** 1Department of Orthopedics, 960th Hospital of PLA, Jinan 250031, China; 2Department of Orthopedics, 80th Group Army Hospital of PLA, Weifang 261000, China

**Keywords:** spondylodiscitis, clinical outcome, negative pressure wound therapy(NPWT), spinal infection

## Abstract

**Purpose:** To investigate the effectiveness and safety of negative-pressure wound therapy (NPWT) in treating primary spinal infections. **Methods:** Patients who underwent surgical treatment for primary spinal infection between January 2018 and June 2021 were retrospectively evaluated. They were divided into two groups based on the type of surgery: one that underwent negative-pressure wound therapy (NPWT) and another that underwent conventional surgery (CVSG-Posterior debridement, bone grafting, fusion, and internal fixation in one stage). The two groups were compared in terms of the total operation time, total blood loss, total postoperative drainage, postoperative pain score, time for the postoperative erythrocyte sedimentation rate (ESR) and C-reactive protein (CRP) to return to normal, postoperative complications, treatment time, and recurrence rate. **Results:** A total of 43 cases of spinal infection were evaluated, with 19 in the NPWT group and 24 in the CVSG group. The NPWT group had a superior postoperative drainage volume, antibiotic use time, erythrocyte sedimentation rate and CRP recovery times, VAS score at 3 months after the operation, and cure rate at 3 months after operation compared with the CVSG group. There were no significant variations in the total hospital stay and intraoperative blood loss between the two groups. **Conclusions:** This study supports the use of negative pressure in the treatment of a primary spinal infection and indicates that it has a notably better short-term clinical effect than conventional surgery. Additionally, its mid-term cure rate and recurrence rate are more desirable than those of conventional treatments.

## 1. Introduction

The number of spinal infections is gradually increasing [1]. Fortunately, most cases can be treated conservatively, with a reported cure rate of 72% [2]. However, successful conservative treatment necessitates timely diagnosis, treatment, and the absence of abscess formation and neurological symptoms. In cases where conservative treatment fails to produce results or dead bone formation occurs, surgical intervention is necessary. Unfortunately, due to the gradual onset and unique clinical signs of spinal infection, definitive diagnosis is often delayed by an average of two to six months [3], often leaving patients in a critical state when they seek medical attention.

Owing to the difficulty of achieving complete debridement with minimally invasive spine surgery, open debridement and internal fixation remain the standard surgical treatments for spinal infections. Conventional open surgery, however, causes increased trauma, a longer operation time, and more blood loss, which may be too much for patients in a poor condition to tolerate. Single open spinal fusion surgery for spinal infections carries a greater risk of recurrence, thus necessitating the development of a minimally invasive spinal surgery method with complete debridement for the surgical treatment of spinal infections.

Negative-pressure wound therapy (NPWT) has been widely recognized for its therapeutic benefits to soft tissue infections, limb bone infections, and chronic refractory wounds since its introduction in the 1990s. Numerous studies have been conducted to assess the efficacy of NPWT in the treatment of bone and soft tissue infections. Luo et al. reported the successful treatment of 12 patients with sacroiliac joint tuberculosis (TB) using NPWT, with an improved hospital stay and C-reactive protein (CRP) and erythrocyte sedimentation rate (ESR) decrease rates compared with conventional treatments [4]. Additionally, Mueller et al. [5] demonstrated, in a prospective study, that NPWT significantly reduces the incidence of postoperative infection following spine surgery compared with conventional wound dressings. Zhang et al. [6] also found that NPWT may reduce the operation time and hospitalization time of patients with fracture infection, accelerate wound healing, and decrease the risk of postoperative infection and lower-extremity deep vein thrombosis. Duan et al. [7] examined the efficacy of thorough debridement and NPWT in the treatment of soft tissue infection and found that NPWT was successful at eliminating necrotic tissue pieces from wounds, draining wound exudates continuously and completely, and encouraging wound healing. Ma et al. [8] studied 73 calcaneal fracture patients with serious infections and found that NPWT treatment was superior to conventional treatments in terms of the wound healing, bacterial clearance, and infection healing times. However, there have been no reports of NPWT being used to treat spinal infections.

A retrospective study was conducted to evaluate the efficacy and safety of negative-pressure wound therapy (NPWT) as an adjunct to surgical debridement in spinal infections, and the results were compared with those of a control group of patients who underwent conventional surgery and open packing during the same period.

## 2. Patients and Methods

### 2.1. Study Subjects

This study included patients who were admitted between 1 January 2018 and 31 June 2021, and had been clinically diagnosed with lumbar infection without spinal cord injury. Post-surgery, a single type of drainage (negative-pressure or traditional catheter drainage) was applied. The Ethics Committee of the 960th Hospital of the PLA approved this investigation (approval number: 2022106). Exclusion criteria included receiving both negative-pressure drainage and traditional drainage, having concomitant spinal deformity, being diagnosed with sepsis, and having a postoperative observation of less than one year.

In this retrospective study, the grouping method was based on staged negative-pressure drainage (NPWT) or traditional primary debridement and bone graft fusion internal fixation (CVSG). A comparison of gender, age, data related to the surgery, and postoperative monitoring indicators was conducted on two groups of patients who had all received surgical treatment and were observed for more than a year (Figure 1).

This study included a total of 43 individuals, consisting of 26 males and 17 females, with an average age of 57.33 years. A total of 19 patients tested positive for bacterial cultures in the two groups, all of whom were administered antibiotics sensitive to their disease, with the traditional drainage group having 3 cases of Staphylococcus aureus, 2 cases of coagulase-negative staphylococcus, 1 case of tuberculosis bacillus, 1 case of Escherichia coli, 1 case of Klebsiella pneumoniae, and 1 case of an unidentified Gram-negative bacillus. Meanwhile, the negative pressure drainage group had 1 case of Staphylococcus aureus, 2 cases of coagulase-negative staphylococcus, 1 case of Escherichia coli, 1 case of Klebsiella pneumoniae, 2 cases of Streptococcus vivostreus, and 3 cases of an unidentified Gram-negative bacillus.

We selected the most effective antibiotics to use based on drug sensitivity tests. If this was not feasible, then antibiotics with a wide spectrum of activity were utilized. Intravenous therapy was used during hospitalization, then changed to oral medication after discharge until clinical recovery. The clinical treatment criteria included the following: symptoms, body temperature, and erythrocyte sedimentation gradually returning to normal; lower-back pain symptoms significantly improving; imaging showing that the abscess had disappeared, intervertebral calcification, resorption of dead bone, fibrosis or ossification, and bone bridge formation; and MRI revealing that the inflammatory changes had disappeared and the signals of the vertebral body and surrounding tissues were normal. It was expected that the patient would be able to maintain the aforementioned standards while living and working as normal for a period of six months.

### 2.2. Surgical Technique

The surgical methods in the NPWT group were: a one-stage debridement with negative-pressure drainage and internal fixation, and a two-stage iliac bone grafting. The negative-pressure sponge (Figure 2) should be inserted into the affected intervertebral space, and negative-pressure suction should be applied for a period of 1–2 weeks post-surgery. The surgical methods in the CVSG group were as follows: posterior debridement, bone grafting, fusion, and internal fixation in one stage. All patients in the CVSG group underwent conventional drainage after surgery.

### 2.3. Data Collection

The total operation time, intraoperative blood loss, postoperative drainage volume, duration of hospitalization, and visual analog scale (VAS) were all documented for each patient in both groups. Additionally, CRP, ESR, and blood routine were regularly monitored during the hospital stay. Upon discharge, the patients were followed up on a bi-weekly basis for the same indicators. Follow-up continued after the patients left the hospital, with outpatient visits conducted on the 1st, 3rd, 6th, and 12th months post-surgery. This enabled the medical team to analyze data such as the time of antibiotic use and when the ESR and CRP returned to normal, and to assess the patient’s overall condition.

### 2.4. Statistical Analysis

The SPSS 22.0 software package was utilized to analyze the data, which were expressed as the mean ± standard deviation. Student’s *t*-test and Pearson’s chi-squared analysis were conducted for the continuous and categorical variables, respectively. Statistical significance was set at an alpha level of less than 0.05.

## 3. Results

The average duration of follow-up was 33.28 ± 11.86 months, with the range being between 16 and 56 months. The NPWT group consisted of 19 patients, 12 males and 7 females, with an average age of 59.00 ± 9.68 years. The infection sites included 3 cases of thoracic spine, 1 case of thoracolumbar spine, and 15 cases of lumbar spine. The CVSG group was made up of 24 patients, 14 males and 10 females, with an average age of 56.00 ± 16.30 years. The infection sites were 3 cases of thoracic spine, 1 case of thoracolumbar spine, and 20 cases of lumbar spine (see Table 1). There was no significant difference in age between the two groups. The baseline characteristics are presented in Table 1.

The results of the comparison between the NPWT and CVSG groups in terms of surgical outcomes are presented in Table 2. The NPWT group had an average hospital stay of 45.16 ± 16.01 days, total surgery duration of 99.74 ± 43.12 min, total blood loss of 105.79 ± 98.11 mL, and total postoperative drainage volume of 364.47 ± 177.84 mL. The CVSG group had an average hospital stay of 34.92 ± 18.32 days, total operation duration of 126.92 ± 59.42 min, total blood loss of 107.50 ± 153.99 mL, and total postoperative drainage volume of 103.96 ± 84.30 mL. There was no significant difference between the two groups in terms of the total operation time, total blood loss, or hospital stay. However, the amount of postoperative drainage was significantly higher in the NPWT group compared with the CVSG group (*p* < 0.05).

The NPWT group showed significantly better results than the CVSG group in terms of the antibiotic usage duration (4.13 ± 1.38 months vs. 5.60 ± 2.28 months), time for ESR to return to normal (53.11 ± 33.57 days vs. 91.64 ± 50.07 days), time for CRP to return to normal (47.11 ± 30.67 days vs. 74.23 ± 45.87 days), VAS score at 3 months after surgery (1.47 ± 0.77 points vs. 2.50 ± 1.38 points), and cure rate at 3 months after surgery (13 cases vs. 0 cases). Moreover, the NPWT group had a significantly higher mid-term cure rate and lower recurrence rate 12 months after surgery than the CVSG group (Table 3 and Table 4).

## 4. Discussion

The prevalence of spinal infections is on the rise due to the accelerated aging process, improved diagnosis, and larger susceptible population. Drug-resistant strains are two and a half times more common than in the general population [9]. Treatment of these infections is difficult, but an early diagnosis can be addressed with conservative measures, such as antibiotics, bed rest, and a spine fixation brace [10]. Typically, one to two weeks of bed rest is recommended, but up to six weeks may be necessary if the infection has spread extensively and compromised spinal stability [10]. However, prolonged immobility in the elderly carries a high risk of developing pressure ulcers, pulmonary embolism, and respiratory infections [11]. Antibiotics are still a necessary part of treatment, despite the fact that they are not as successful as other bone infection treatments [10]. This is due to the difficulty of locating pathogenic microorganisms in spinal infections in comparison with other kinds of bone infections [12], the absence of blood flow to the intervertebral disc tissue, and the quick propagation of infection. In certain cases of spinal infection with co-morbidities, conservative treatment is not successful in alleviating neurological symptoms, lumbar instability, kyphosis, spinal abscess, infection of more than four vertebral bodies, and infection of the intervertebral disc [13]. In these cases, surgery is the only viable option. Approximately half of all patients with spinal infections require surgery [14,15]. Studies have demonstrated that the risk of recurrence of spinal infection can be drastically reduced if the infected area is completely eliminated [16,17]. As spinal infection is located in a specific area of the body, it cannot be treated with the same extensive debridement as bone infection in extremities. Furthermore, the recurrence rate of a single operation is often quite high.

NWPT is effective in extracting exudate, pus, and other substances through the application of negative pressure. Furthermore, NPWT is able to break down large, soft tissues and secretions into smaller particles, which can then be aspirated. An even suction pressure is applied to the surface of polyvinyl alcohol (PVA), forming a drainage system that extends in all directions. Sequestrum, pus, and bloodless tissues are known to have the lowest concentrations of drugs in the bloodstream, making them more susceptible to the development of drug-resistant bacteria and bacterial biofilms, which are major contributors to recurrent infections [18,19,20]. PVA has excellent water permeability and does not impede the flow of liquids and tiny particles, thus allowing negative-pressure wound therapy (NPWT) to drain the water from the bacterial biofilm, leading to necrosis of the bacteria in the biofilm and eliminating the growth environment. NPWT can simultaneously reduce the bacterial load and minimize the dead space at the infection site. Furthermore, the drainage tube is encased in PVA foam, which prevents contact between the tube and the surrounding tissues and organs, thus avoiding inhalation of the surrounding tissue, which could cause ischemia, necrosis, and local hemorrhage. PVA can also help to protect the wound site from external sources of infection and may promote wound healing [21].

Presently, the traditional surgical approach to spinal infection is debridement, drainage, and bone grafting and internal fixation for patients with spinal instability. However, this type of operation can be lengthy and cause a great deal of trauma and bleeding, as well as potentially not completely debriding the area. Additionally, individuals with preexisting medical issues have a significantly higher rate of severe complications and mortality following the procedure compared with individuals without preexisting conditions [22,23]. In addition, due to the special location of spinal infection, extensive debridement, like that of the fourth limb bone infection, cannot be taken. No studies have been reported on the use of NPWT for the treatment of primary spinal infection, but we have seen successful results with its application in spondylodiscitis. Taking inspiration from the success of NPWT in treating bone and soft tissue infections, postoperative spinal infections, etc., we introduced the NPWT device into the intervertebral space for the treatment of primary spinal infections with the aim of eliminating the foci and preventing recurrence. Compared with conventional methods, the NPWT group showed significant improvements in the postoperative drainage volume, ESR and CRP recovery times, and antibiotic use duration. Three months after surgery, the short-term cure rate was significantly higher in the NPWT group. Additionally, all patients in the NPWT group were cured without recurrence 1 year after surgery, which was significantly higher than that of the CVSG group. However, this conclusion requires further research with a larger sample size to be confirmed.

The treatment of spinal infection involves two distinct processes: controlling the infection and repairing the tissue defect at the site of infection, which includes bone and soft tissue repair. Conventional methods typically address these two processes concurrently. However, the use of negative-pressure wound therapy (NPWT) temporally separates infection control and tissue repair, allowing for tissue repair to occur after the infection has been cleared. This technique has been shown to increase the cure rate and decrease the recurrence rate of spinal infections. Additionally, NPWT reduces local edema and bacterial loads, destroys the local biofilm, and attracts pus and small necrotic tissue out of the body, improving local blood circulation and blood concentration at the site of infection. As such, NPWT is superior to conventional surgical techniques for treating primary spinal infections, and the use of a second-stage iliac bone transplant further reduces the surgical difficulty and danger of anesthesia.

## 5. Conclusions

This study reveals that NPWT technology has a notably better short-term clinical effect than conventional surgery, and its mid-term cure rate and recurrence rate are more desirable than those of conventional treatments. The efficacy of NPWT in treating spinal infections in the short and medium terms is clear, and its use is expected to become commonplace in clinical settings. This research is limited by the need to include a larger sample size and to conduct a study across multiple sites.

## Figures and Tables

**Figure 1 jpm-13-00162-f001:**
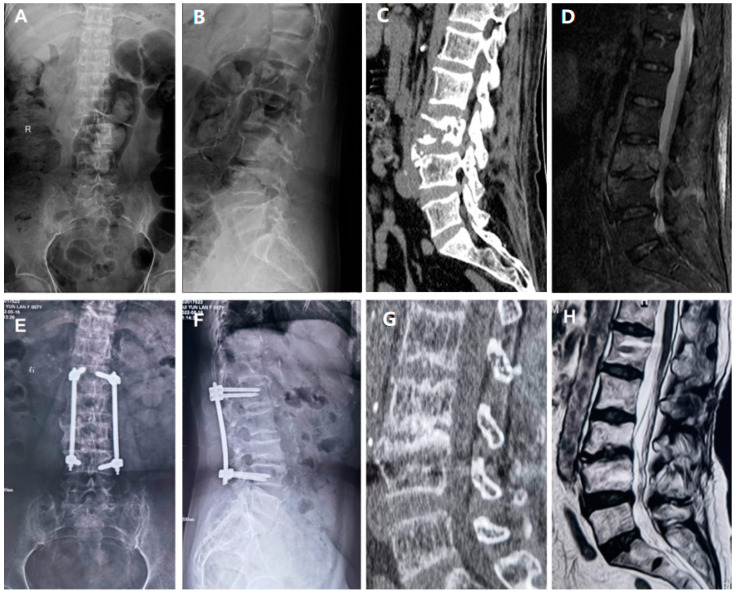
Illustrative Case. A 54-year-old female patient was admitted to the hospital with a diagnosis of lumbar spine infection. Under general anesthesia, posterior debridement of lumbar spine lesions and percutaneous screw internal fixation with VSD suction was performed on 18 July 2019. On 25 July 2019, further lumbar spine lesion removal (L2/3) and VSD negative-pressure drainage were conducted. On 1 August 2019, lumbar spine lesion removal, lumbar intervertebral space bone grafting, and VSD negative-pressure drainage were conducted. (**A**,**B**) An admission X-ray showing the destruction of the vertebral bone. (**C**) Destruction of the L3 vertebral body seen on the admission CT. (**D**) Magnetic resonance imaging on admission showing signs of infection in the L3/4 intervertebral space. (**E**,**F**) X-rays 1 year after surgery showing an L3/4 fusion. (**G**) Fusion of the vertebral body seen on CT a year after surgery. (**H**) MRI showing the disappearance of signs of infection one year after surgery.

**Figure 2 jpm-13-00162-f002:**
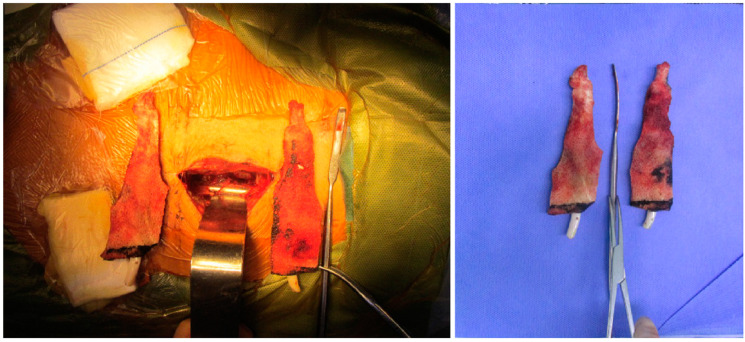
VSD sponge placed in the intervertebral space (after removal).

**Table 1 jpm-13-00162-t001:** Characteristics of the study subjects.

Group	Male	Female	Age	Infection Location			Involved Segments	Fixation Devices		
				Thoracic	Thoracolumbar	Lumbar		Anterior	Posterior	Unfixed
NPWT	12	7	59.00 ± 9.68	3	1	15	1.21	1	13	5
CVSG	14	10	56.00 ± 16.30	3	1	20	1.13	2	15	7

**Table 2 jpm-13-00162-t002:** Comparisons of NPWT and non-NPWT groups’ surgical outcomes.

Group	Duration of Hospitalization (d)	Total Operation Time (min)	Blood Loss (mL)	Volume of Drainage Fluid (mL)
NPWT	45.16 ± 16.01	99.74 ± 43.12	105.79 ± 98.11	364.47 ± 177.84
CVSG	34.92 ± 18.32	126.92 ± 59.42	107.50 ± 153.99	103.96 ± 84.30
T value	1.92	−1.67	−0.042	6.35
*p* value	0.061	0.102	0.967	0.000

**Table 3 jpm-13-00162-t003:** Comparison of surgical conditions between NPWT and CVSG groups.

Antibiotic Usage Duration (m)		Time for ESR to Return to Normal (d)	Time for CRP to Return to Normal (d)	VAS Score 3 Months after Surgery
NPWT	4.13 ± 1.38	53.11 ± 33.57	47.11 ± 30.67	2.50 ± 1.38
CVSG	5.60 ± 2.28	91.64 ± 50.07	74.23 ± 45.87	2.50 ± 1.38
T value	−2.48	−2.85	−2.19	−2.89
*p* value	0.017	0.007	0.035	0.006

**Table 4 jpm-13-00162-t004:** Comparisons of NPWT and CVSG groups’ surgical outcomes.

	Cured 3 Months after Surgery		Cured 12 Months after Surgery		Recurrence 12 Months after Surgery	
	Yes	No	Yes	No	Yes	No
NPWT	13	6	19	0	0	19
CVSG	8	16	19	5	5	19
χ^2^ value	5.23		4.48		4.48	
*p* value	0.022		0.034		0.034	

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
