# Peer review of "A Comparison of Negative Pressure and Conventional Therapy in Spine Infections: A Single-Center Retrospective Study"

_jpm, 2023, doi:10.3390/jpm13020162_

Round 1

Reviewer 1 Report

This paper addresses a very important issue in orthopedic surgery, the surgical treatment of spondylodiscitis.

Line 37: The authors mentioned the issue of abscess and neurological deterioration, but also failed to mention the issue of sequestration and, the lack of response to treatment

Line 43:” conventional open surgery has greater trauma”, should be: “conventional open surgery causes greater trauma”.

Line 72: The authors mentioned that: “According to different surgical methods, they were divided into the NPWT group and the CVSG group”. The exact criteria for assignment should be detailed.

Line 74: The authors mentioned that “The gender, age, surgery-related data and postoperative 74 monitoring indicators of the two groups of patients were all comparable”. Data should be detailed in a table.

Lines 77-79. Bacterial resistance should be detailed.

Line 118: Follow up periods should be detailed including mean, SD and range.

Line 131: “The CVSG group had an average hospital stay of 34.9218.32 days” number is unclear.

Line 132: “total blood loss of 107.50±153.99 ml”. Did some patients gained blood during surgery?

Antibiotic regimens should be detailed.

Table 3: “he time for ESR to return to normal (d)”. Should be corrected.

The authors did not discuss the limitations of the study, especially the fact that it is almost impossible to get conclusion from a retrospective study without randomization.

Author Response

We are grateful for the constructive feedback provided by the editors and reviewers. We have taken the time to address each of their comments and have made the necessary revisions. Please see the responses for further details. Should there be any discrepancies, we are more than willing to make further amendments. 

Thank you for giving us the opportunity to submit a revised manuscript, we appreciate the time and effort of the reviewers. The following is our point-to-point response to the reviewers:

Reviewer #1

This paper addresses a very important issue in orthopedic surgery, the surgical treatment of spondylodiscitis.

  1. Line 37: The authors mentioned the issue of abscess and neurological deterioration, but also failed to mention the issue of sequestration and, the lack of response to treatment.

Response: The reviewer's opinion is accurate; if dead bone formation or conservative treatment is unsuccessful, surgical treatment is necessary. This has been updated in the article.

  1. Line 43:” conventional open surgery has greater trauma”, should be: “conventional open surgery causes greater trauma”.

Response: The reviewer's opinion has been taken into consideration and the article has been amended accordingly.

  1. Line 72: The authors mentioned that: “According to different surgical methods, they were divided into the NPWT group and the CVSG group”. The exact criteria for assignment should be detailed.

Response: In this retrospective study, what was the grouping method based on: staged negative pressure drainage surgery(NPWT), or traditional one-stage debridement, bone graft fusion and internal fixation(CVSG).

  1. Line 74: The authors mentioned that “The gender, age, surgery-related data and postoperative monitoring indicators of the two groups of patients were all comparable”. Data should be detailed in a table.

Response: The reviewer's opinion has been taken into consideration and the relevant tables have been amended accordingly.

  1. Lines 77-79. Bacterial resistance should be detailed.

Response: Results from bacterial cultures of 19 patients were positive, and all of them had antibiotics that were sensitive to their condition. We followed the drug susceptibility tests to determine the most effective antibiotics to use.

  1. Line 118: Follow up periods should be detailed including mean, SD and range.

Response: The average duration of follow-up was 33.28 ± 11.86 months, with the range being between 16 and 56 months.

  1. Line 131: “The CVSG group had an average hospital stay of 34.9218.32 days” number is unclear.

Response: The reviewer's opinion has been taken into consideration and the article has been amended accordingly.

  1. Line 132: “total blood loss of 107.50±153.99 ml”. Did some patients gained blood during surgery?

Response: During the operation, no patients gained blood.

  1. Antibiotic regimens should be detailed.

Response: For patients with a positive culture, antibiotics should be selected to match their sensitivities; if this is not feasible, then antibiotics with a wide spectrum of activity should be utilized. During hospitalization, intravenous therapy should be administered, and after discharge, oral medication should be taken.

  1. Table 3: “he time for ESR to return to normal (d)”. Should be corrected.

Response: The reviewer's opinion has been taken into consideration and the article has been amended accordingly.

  1. The authors did not discuss the limitations of the study, especially the fact that it is almost impossible to get conclusion from a retrospective study without randomization.

Response: The reviewer is correct in their opinion that a retrospective study contains certain biases. We are in the process of registering prospective single-arm studies, which are also retrospective with conventional surgery. In this group of cases, the preoperative comprehensive situation of patients receiving traditional surgery is better than that of the NPWT group, yet the final treatment result of the NPWT group is better than that of the CVSG group. Therefore, even though this is a retrospective study, it does not negate the accuracy of the conclusion.

Reviewer 2 Report

A comparison of negative pressure and conventional therapy in spine infections: a single-center retrospective study

This manuscript discusses clinical findings for a spine infection, comparing a control group and a group treated using a negative pressure system.

Overall Comments: This is a very interesting topic. The finding that the wound vacuum treatment seems to effective is highly interesting but there are a number of serious concerns about this study. A number of important pieces of information are missing from this manuscript. First, the study does not mention ethics oversight, and initially it is described as retrospective then later as prospective.   Another concern is that it is not clear whether the patient groups were truly comparable, as factors related to wound healing are not presented. For example, no comorbid conditions are described, no measure of extent of infection is provided, and the infecting species was not documented for many cases (and infection type was not compared between groups).  Lastly, the antibiotic treatment is not well described.  Based on these factors the manuscript is currently not suitable for publication.

Specific Comments:

Line 35 – Please be more specific in regard to what is trending upward: number of cases? Location? Risk of infection?

Line 44: The term “afford” might suggest to the readers that financial factors play a role. The term “tolerate” might be better here.

Line 51 and other lines: When a term is first used, please spell out the entire term then provide the initialization (example TB on this line and same comment for other lines in this manuscript).

Line 52: please refrain from using “etc” in a scholarly manuscript. It is imprecise and informal. It would be better to list all items or indicate that additional items are possible.

Line 65: retrospective trial is not a recognized study design, “trial” suggests prospective

Where is the statement in regard to IRB/ethics review?

Methods: Please include inclusion and exclusion criteria for the study; please indicate which species of bacteria were present in each group; please provide further patient information to assure the reader that the groups were comparable (example: number or type of comorbid conditions, degree/spread of infection, WBC counts, or similar).

It is concerning that there is no bacteria information for so many of the subjects and unclear how infection was verified for these cases. Please provide more explanation.

Line 82: The text says the groups were composed “according to surgical method” but I am not clear on whether this just means that they were wound vac/conventional treatment or was there some other difference in terms of the surgery?

Line 104: “prospectively” what does this mean here – were the patients tracked after the researchers identified them following surgery?  This is very strange to me.

Line 113: Did the authors test the data for normality prior to using t-test and reporting means?

Table 1 should include more details to help with the comparison. What fixation devices? How long is the incision? How much volume/area is involved in infection?

Line 131: typo

Line 139: What antibiotic? How did you know it was appropriate for the species (if no info on the species as was the case for many patients)? What dosage/delivery method?

Line 141: What is your “cure criteria”

Discussion: I am not providing a detailed review of the Discussion, because there are so many issues with the earlier parts of the paper. In general the authors should take care to update the discussion with limitations, provide appropriate references for statements, and revise the discussion to improve English language clarity.

Author Response

We are grateful for the constructive feedback provided by the editors and reviewers. We have taken the time to address each of their comments and have made the necessary revisions. Please see the responses for further details. Should there be any discrepancies, we are more than willing to make further amendments. 

Thank you for giving us the opportunity to submit a revised manuscript, we appreciate the time and effort of the reviewers. The following is our point-to-point response to the reviewers:

Reviewer #2

This manuscript discusses clinical findings for a spine infection, comparing a control group and a group treated using a negative pressure system.

Overall Comments: This is a very interesting topic. The finding that the wound vacuum treatment seems to effective is highly interesting but there are a number of serious concerns about this study. A number of important pieces of information are missing from this manuscript. First, the study does not mention ethics oversight, and initially it is described as retrospective then later as prospective.   Another concern is that it is not clear whether the patient groups were truly comparable, as factors related to wound healing are not presented. For example, no comorbid conditions are described, no measure of extent of infection is provided, and the infecting species was not documented for many cases (and infection type was not compared between groups).  Lastly, the antibiotic treatment is not well described.  Based on these factors the manuscript is currently not suitable for publication.

Response: Your efforts have been instrumental in ensuring our articles have a scientific basis. We appreciate your commitment. Ethical oversight and comprehensive antibiotic therapy have been incorporated into the article. The reviewer is correct in their opinion that a retrospective study contains certain biases. We are in the process of registering prospective single-arm studies, which are also retrospective with conventional surgery. In this group of cases, the preoperative comprehensive situation of patients receiving traditional surgery is better than that of the NPWT group, yet the final treatment result of the NPWT group is better than that of the CVSG group. Therefore, even though this is a retrospective study, it does not negate the accuracy of the conclusion.

Specific Comments:

  1. Line 35 – Please be more specific in regard to what is trending upward: number of cases? Location? Risk of infection?

Response: The number of spinal infections is gradually increasing. The article has been revised.

  1. Line 44: The term “afford” might suggest to the readers that financial factors play a role. The term “tolerate” might be better here.

Response: The reviewer's comments are correct and it has been modified to tolerate.

  1. Line 51 and other lines: When a term is first used, please spell out the entire term then provide the initialization (example TB on this line and same comment for other lines in this manuscript).

Response: The reviewer's comments are correct, and it has been modified in the manuscript.

  1. ine 52: please refrain from using “etc” in a scholarly manuscript. It is imprecise and informal. It would be better to list all items or indicate that additional items are possible.

Response: The reviewer's comments are correct, and "etc" has been taken out.

  1. Line 65: retrospective trial is not a recognized study design, “trial” suggests prospective

Response: The reviewer's comments are correct, and it has been modified in the manuscript.

  1. Where is the statement in regard to IRB/ethics review?

Response: This investigation was authorized by the Ethics Committee of the 960th Hospital of the PLA (approval number: 2022106).

  1. Methods: Please include inclusion and exclusion criteria for the study; please indicate which species of bacteria were present in each group; please provide further patient information to assure the reader that the groups were comparable (example: number or type of comorbid conditions, degree/spread of infection, WBC counts, or similar).

Response: The reviewer's comments are correct, and it has been modified in the manuscript.

Patients included in this study had an admission time between January 1, 2018 and June 31, 2021, were clinically diagnosed with lumbar infection without spinal cord injury, and had a single type of drainage (negative pressure or traditional catheter drainage) applied post-surgery. Exclusion criteria included receiving both negative pressure drainage and traditional drainage, having concomitant spinal deformity, being diagnosed with sepsis, and having a postoperative follow-up of less than one year.

  1. It is concerning that there is no bacteria information for so many of the subjects and unclear how infection was verified for these cases. Please provide more explanation.

Response: 19 patients were identified to have bacterial cultures, with the traditional drainage group having 3 cases of Staphylococcus aureus, 2 cases of coagulase negative staphylococcus, 1 case of tuberculosis bacillus, 1 case of Escherichia coli, 1 case of Klebsiella pneumoniae, and 1 case of an unidentified Gram-negative bacillus. Meanwhile, the negative pressure drainage group had 1 case of Staphylococcus aureus, 2 cases of coagulase negative staphylococcus, 1 case of Escherichia coli, 1 case of Klebsiella pneumoniae, 2 cases of Streptococcus vivostreus, and 3 cases of an unidentified Gram-negative bacillus.

  1. Line 82: The text says the groups were composed “according to surgical method” but I am not clear on whether this just means that they were wound vac/conventional treatment or was there some other difference in terms of the surgery?

Response: Patients were classified into two groups, NPWT and CVSG, depending on whether they underwent negative pressure wound therapy.

  1. Line 104: “prospectively” what does this mean here – were the patients tracked after the researchers identified them following surgery? This is very strange to me.

clerical error,not “prospectively”.

Response: The reviewer's comments are correct, and "prospectively" has been taken out.

  1. Line 113: Did the authors test the data for normality prior to using t-test and reporting means?

Response: Before conducting a t-test and reporting means, we assess the data for normality.

  1. Table 1 should include more details to help with the comparison. What fixation devices? How long is the incision? How much volume/area is involved in infection?

Response: The reviewer's comments are correct, and it has been modified in the manuscript.

  1. Line 131: typo

Response: The reviewer's comments are correct, and it has been modified in the manuscript.

  1. Line 139: What antibiotic? How did you know it was appropriate for the species (if no info on the species as was the case for many patients)? What dosage/delivery method?

Response: For patients with a positive culture, antibiotics should be selected to match their sensitivities; if this is not feasible, then antibiotics with a wide spectrum of activity should be utilized. During hospitalization, intravenous therapy should be administered, and after discharge, oral medication should be taken.

  1. Line 141: What is your “cure criteria”

Response: The relevant information is located between lines 108 and 115.

  1. Discussion: I am not providing a detailed review of the Discussion, because there are so many issues with the earlier parts of the paper. In general the authors should take care to update the discussion with limitations, provide appropriate references for statements, and revise the discussion to improve English language clarity.

Response: The reviewer's comments are correct, and it has been modified in the manuscript.

Round 2

Reviewer 1 Report

No further comments

Author Response

We are grateful to the reviewer for their invaluable contribution to our article. Your assistance has enabled us to make our article more accurate and comprehensive. We are thankful for reviewer #1's approval of our article, allowing us the chance for other scholars to evaluate our work.

Reviewer 2 Report

Parts of the manuscript are improved, but some parts seem to have been hastily put together with grammatical errors.  A revision is again required to fix these issues. 

Abstract: The conclusion statement here is too strongly worded. A retrospective review cannot “prove” safety as it has inherent biases and is non-randomized.  I suggest wording like “this study supports the use of negative pressure” as this is more in line with the strength of the study.

Methods: This section is poorly written, with sections still appearing to suggest a prospective study.  It describes techniques and care using present tense, but the study was retrospective so these should be past tense in my opinion. It should also clarify that these are the typical practices for the facility, since the study did not control treatment practices (as it was a retrospective review). 

Page 7 “what was the grouping based on” seems to be out of place here

Page 7 bottom: “all of them had antibiotics..” grammar issue/poorly worded sentence; also there is also a problem with the numbers here with a different number on page 8?

Page 8: “We followed the” implies prospective – should be worked in past tense and generalized to be consistent with the study design

Page 9 top: This paragraph is a repeat from earlier. It should be removed.

The results are improved in this version.

The Discussion is also greatly improved.

The Conclusions are reasonable.

Author Response

We are grateful for the comments from reviewer #2, which have enabled us to make our article more scientific and comprehensive. We have taken the reviewer's suggestions into account and have made the necessary modifications. If there are any further areas that need improvement, we are ready to make more changes.
